# Self-Reported Long COVID and Its Association with the Presence of SARS-CoV-2 Antibodies in a Danish Cohort up to 12 Months after Infection

Kamille Fogh,[a,b,g] Tine Graakjær Larsen,[c] Cecilie B. Hansen,[d,g] Rasmus B. Hasselbalch,[a,b,g] Alexandra R. R. Eriksen,[a,b,g] Henning Bundgaard,[e,g] Ruth Frikke-Schmidt,[f,g] Linda M. Hilsted,[g] Lars Østergaard,[h,i] Isik S. Johansen,[j,k] Ida Hageman,[l] Peter Garred,[d,g] Kasper Iversen[a,b,g]

[a]Department of Cardiology, Copenhagen University Hospital, Herlev and Gentofte, Denmark
[b]Department of Emergency Medicine, Copenhagen University Hospital, Herlev and Gentofte, Denmark
[c]Department of Infectious Disease Epidemiology and Prevention, Statens Serum Institut, Copenhagen, Denmark
[d]Laboratory of Molecular Medicine, Department of Clinical Immunology, Rigshospitalet, Copenhagen, Denmark
[e]Department of Cardiology, Copenhagen University Hospital, Rigshospitalet, Copenhagen, Denmark
[f]Department of Clinical Biochemistry, Copenhagen University Hospital, Rigshospitalet, Copenhagen, Denmark
[g]Department of Clinical Medicine, Copenhagen University Hospital, Faculty of Health and Medical Sciences, Copenhagen, Denmark
[h]Department of Infectious Diseases, Aarhus University Hospital Aarhus, Denmark
[i]Department of Clinical Medicine, Aarhus University, Aarhus, Denmark
[j]Department of Infectious Diseases, Odense University Hospital, Odense, Denmark
[k]Department of Clinical Research, University of Southern Denmark, Odense, Denmark
[l]Mental Health Services, Copenhagen, Denmark

**ABSTRACT** The majority of long coronavirus disease (COVID) symptoms are not specific to COVID-19 and could be explained by other conditions. The present study aimed to explore whether Danish individuals with a perception that they suffer from long COVID have antibodies against the nucleocapsid antigen, as a proxy for detecting previous infection. The study was conducted in February and March 2021, right after the second surge of the COVID-19 pandemic in Denmark. All members of the social media group on Facebook "Covidramte med senfølger" ("long COVID sufferers") above the age of 17 years and living in Denmark were invited to participate in a short electronic questionnaire about long COVID risk factors and symptoms. The severe acute respiratory syndrome coronavirus 2 (SARS-CoV-2) nucleocapsid (N) protein was detected in blood samples as a proxy for natural SARS-CoV-2 infection. The final study population comprised 341 participants (90.6% females) who completed blood sampling and answered the questionnaire. A total of 232 (68%) were seropositive (median age, 49.5 years; interquartile range [IQR], 41 to 55 years; 90.1% females). There was no significant difference between sexes and serostatus. Seronegative and seropositive individuals had a similar burden of symptoms that could be attributed to long COVID. Time since perceived COVID-19 was significantly longer in the group of seronegative individuals than the seropositive ones ($P < 0.001$). This study suggests that long-COVID sufferers are mostly women and showed that a third of the participants did not have detectable anti-N-protein antibodies. It emphasizes the importance of early confirmation of COVID-19, as this study indicates an overlap between long-COVID symptoms and symptoms that are possibly of another origin.

**IMPORTANCE** This cohort study included questionnaire data as well as anti-nucleocapsid antibody analysis, allowing us to determine whether participants were seropositive due to vaccination or natural infection. The study emphasizes the importance of early confirmation of COVID-19, as antibodies recede with time, and it indicates an overlap between long COVID symptoms and symptoms possibly of another origin.

Address correspondence to Kamille Fogh, kamille.fogh.01@regionh.dk.

The authors declare no conflict of interest.

**KEYWORDS** long COVID, post-COVID syndrome, prolonged COVID, post-acute-phase COVID-19, PASC, antibodies, SARS-CoV-2 antibodies, nucleocapsid antigen, N protein, SARS-CoV-2

Long-term persistence of symptoms after recovery from severe acute respiratory syndrome coronavirus 2 (SARS-CoV-2) infection is often referred to as long COVID (1). According to the World Health Organization (WHO), a post-coronavirus disease 2019 (COVID-19) condition occurs as a post-acute-phase consequence of COVID-19, usually 3 weeks from onset of COVID-19, with symptoms lasting at least 2 months that cannot be explained by an alternative diagnosis (1). Clinically, it is known as post-acute-phase sequelae of SARS-CoV-2 infection (PASC) (2). The fraction of SARS-CoV-2-infected people developing COVID-19 and experiencing long COVID and the exact nature of the symptoms have not been fully elucidated (3, 4).

Reported symptoms of long COVID are wide-ranging, with the involvement of nearly all organ systems, and with post-COVID fatigue being the most common symptom (5). Post-COVID symptoms have been reported among persons of all ages and in both hospitalized and nonhospitalized COVID-19 patients (6–8) and have been found to occur more frequently with increasing age, higher body mass index, and female sex (9). Long COVID has the potential to reduce quality of life significantly.

Most of the reported long-COVID symptoms are not specific to COVID-19 and could be explained by other conditions. Though focus during the pandemic primarily has been on the direct effects of being infected, society has been exposed to other stressors during the pandemic that could have affected public health independently of infection. For example, during long periods of the pandemic, citizens have been urged to maintain social isolation in order to avoid further spread of SARS-CoV-2. While social isolation has been inconvenient and unfavorable for many people, it is likely that some individuals will have experienced a large psychological impact, leading to an increased risk of impairment to their general mental health (10–12). The present study aimed to explore whether Danish individuals with a perception that they suffer from long COVID have antibodies against the nucleocapsid antigen, as a proxy for detecting previous infection.

## RESULTS

**Study population and characteristics.** Baseline characteristics of the study participants are presented in Table 1. Of the 341 individuals who participated in blood sampling and afterwards filled out the in-depth questionnaire, 32 (9.4%) were men and 309 (90.6%) were women. No significant difference was found in serostatus between males and females ($P = 0.77$), with 71.8% (23 of 32) of men and 67.6% (209 of 309) of women being seropositive. Median age was similar in seronegative and seropositive individuals (49.5 years versus 48 years, respectively; $P = 0.22$). Median body mass index (BMI) was not significantly different ($P = 0.19$) in the two groups. No clear difference in serology was associated with smoking or alcohol consumption.

A total of 232 (68%) participants tested positive for SARS-CoV-2 N-protein antibodies (Table 1), leaving 32% of the participants seronegative. Being seropositive was associated with being employed full- or part-time, though the association was not significant ($P = 0.13$), while individuals who were self-employed, stay-at-home, without a job, on leave from their job, or retired were more frequently observed in the group of seronegative individuals.

**Long-COVID-related symptoms and time since COVID-19 infection.** Almost all symptoms were reported to be worse after than before assumed COVID-19 for both the seropositive and the seronegative groups. Exceptions to this were the symptoms "vomiting" and "diarrhea," which were reported as "better" after assumed COVID-19 (Fig. 1). Only "sensitivity to sound" was significantly worse in the group of seropositive than seronegative individuals ($P < 0.05$); all other symptoms were not significantly different. Figure 2 and Fig. S1 in the supplemental material show the time

**TABLE 1** Baseline characteristics of the study cohort, stratified by seropositivity

| Characteristic | Value for group | | | P value |
| --- | --- | --- | --- | --- |
| | Seropositive | Seronegative | Total | |
| No. of participants | 232 | 109 | 341 | |
| No. of males (%) | 23 (9.9) | 9 (8.3) | 32 (9.4) | 0.772 |
| No. of females (%) | 209 (90.1) | 100 (91.7) | 309 (90.6) | |
| Median age (in yrs) (IQR) | 49.5 (41–55) | 48 (41–54) | 49 (41–55) | 0.217 |
| Median body mass index (IQR) | 25.4 (22.7–29.7) | 24.5 (22.0–28.1) | 25.3 (22.3–29.0) | 0.189 |
| No. with missing data | 2 | 0 | 2 | |
| No. (%) who ever smoked | 39 (16.8) | 21 (19.3) | 60 (17.6) | 0.687 |
| No. (%) reporting alcohol use | | | | |
| Yes | 191 (82.3) | 80 (73.4) | 271 (79.5) | 0.110 |
| No | 40 (17.2) | 29 (26.6) | 69 (20.2) | |
| Do not know/do not wish to answer | 1 (0.4) | 0 (0.0) | 1 (0.3) | |
| No. (%) previously tested | | | | |
| Any test | 223 (96.1) | 100 (91.7) | 323 (94.7) | 0.154 |
| Swab test (PCR) | 216 (93.1) | 98 (89.9) | 314 (92.1) | 0.421 |
| Mean time (in days) since COVID-19 (SD) | 213.9 (134.1) | 317 (96.4) | 248.9 (131.8) | <0.001 |
| Previous antibody test | | | | |
| No. with positive result | 108 (46.6) | 22 (20.2) | 130 (38.1) | |
| No. with negative result | 19 (8.2) | 35 (32.1) | 54 (15.8) | |
| Do not know/missing | 105 (45.3) | 52 (47.7) | 157 (46.0) | <0.001 |
| No. with influenza vaccination in: | | | | |
| 2019 | 50 (21.6) | 17 (15.7) | 67 (19.7) | 0.268 |
| 2020 | 79 (34.1) | 30 (27.5) | 109 (32.0) | 0.280 |
| No. (%) with health status | | | | |
| Excellent | 17 (7.3) | 8 (7.3) | 25 (7.3) | 0.130 |
| Very good | 74 (31.9) | 36 (33.0) | 110 (32.3) | |
| Good | 69 (29.7) | 25 (22.9) | 94 (27.6) | |
| Poor | 60 (25.9) | 26 (23.9) | 86 (25.2) | |
| Very poor | 12 (5.2) | 14 (12.8) | 26 (7.6) | |
| No. (%) COVID-19 vaccinated | 20 (13.4) | 6 (7.8) | 26 (11.5) | 0.300 |
| No. with missing data | 83 | 32 | 115 | |
| No. (%) with work status | | | | |
| Full-time employee | 132 (56.9) | 60 (55.0) | 192 (56.3) | 0.126 |
| Part-time employee | 41 (17.7) | 9 (8.3) | 50 (14.7) | |
| Self employed | 14 (6.0) | 11 (10.1) | 25 (7.3) | |
| Student | 5 (2.2) | 4 (3.7) | 9 (2.6) | |
| Stay-at-home | 2 (0.9) | 1 (0.9) | 3 (0.9) | |
| Out of job | 2 (0.9) | 5 (4.6) | 7 (2.1) | |
| Long-term sick leave/parental leave | 10 (4.3) | 6 (5.5) | 16 (4.7) | |
| Retired | 15 (6.5) | 9 (8.3) | 24 (7.0) | |
| Other | 11 (4.7) | 4 (3.7) | 15 (4.4) | |

since the assumed first day of COVID-19 for seronegative and seropositive individuals. Four individuals were excluded from this part of the analysis; three because they specified a date that was after the date on which they filled out the questionnaire, and one because the individual specified the date on which they filled out the questionnaire.

Time since COVID-19 was significantly longer for the group of seronegative individuals ($P < 0.001$) with a narrow interquartile range compared to the group of seropositive (Fig. S1). The mean time since COVID-19 was 248.9 days (95% confidence interval [CI] = 234.9 to 262.9) for the entire cohort, 317 days (95% CI = 306.7 to 327.3) for

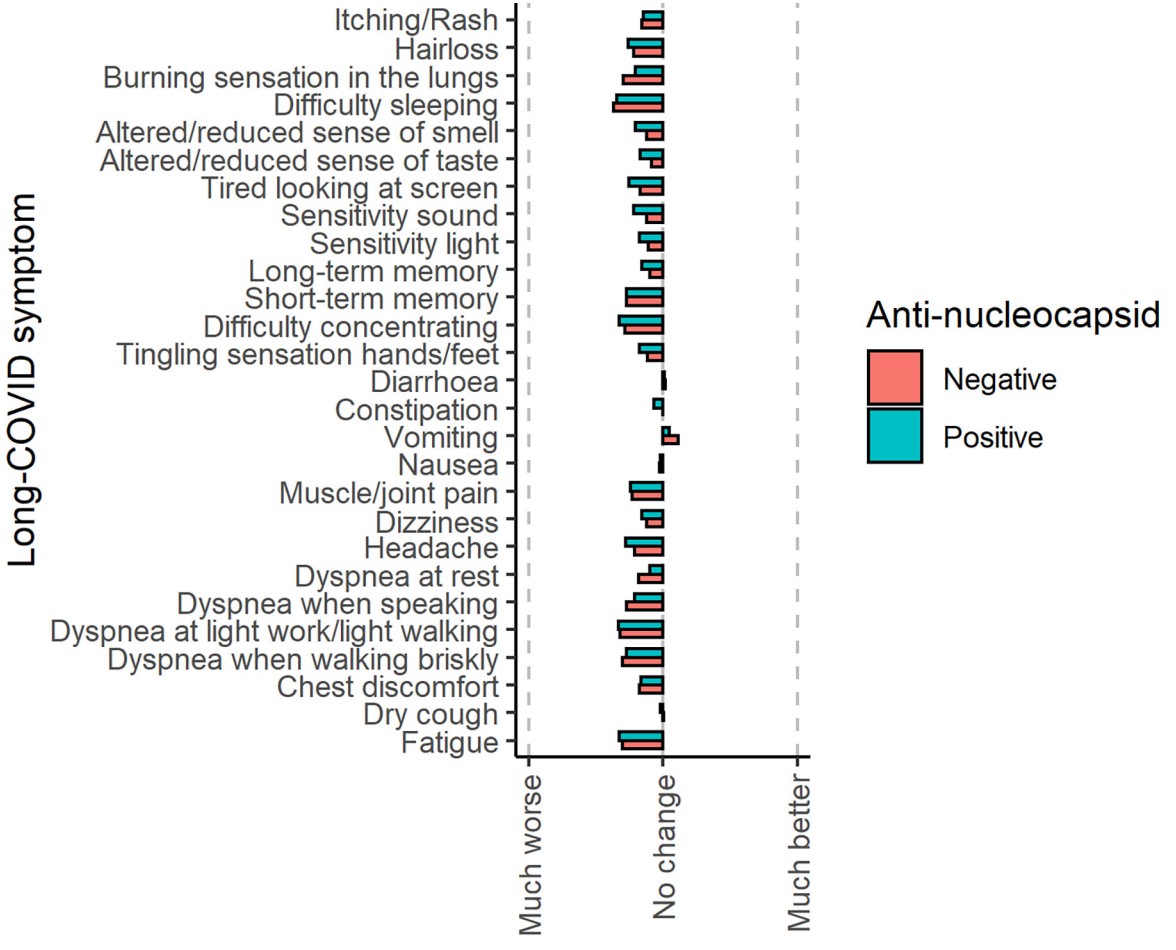

**FIG 1** Mean change in symptoms.

seronegative individuals, and 213.9 days (95% CI = 199.6 to 228.2) for seropositive individuals.

Figure 2 is a histogram showing the dispersion of the "time since the first day of assumed COVID-19 infection" within the cohort. Two peaks appear, the first one around 3 to 4 months after COVID-19, which is dominated by seropositive individuals. The second peak is around 1 year after infection and consists of both seronegative and seropositive individuals.

The individual long-COVID symptoms were not predictive for serostatus in an initial, overfitted logistic regression model where all symptoms were included, as well as age, sex, and time since assumed COVID-19 (Table 2). In the final model, symptom burden was pooled into a "sum of all long-COVID symptoms," which still had no significant predictive value for serostatus. However, in the final model, a 1-day increase in time since assumed COVID-19 was significantly associated with odds of a positive antibody sample decreasing by 1.3%. Therefore, a 1-month (30-day) increase in time since assumed COVID-19 was associated with a reduction in the odds by 32.5%.

**Health impact.** Participants were asked to grade how much of the time in the previous 4 weeks (prior to blood sampling and filling out of the questionnaire) they had felt "calm and peaceful," "full of energy," and "downhearted and blue." Figure 3 shows the proportions of seronegative and seropositive participants who experienced these feelings "none of the time," "a bit of the time," "some of the time," "a great deal of the time," "most of the time" or "all of the time." The experience of these feelings in the 4 weeks prior to blood sampling was similar among seropositive and seronegative individuals.

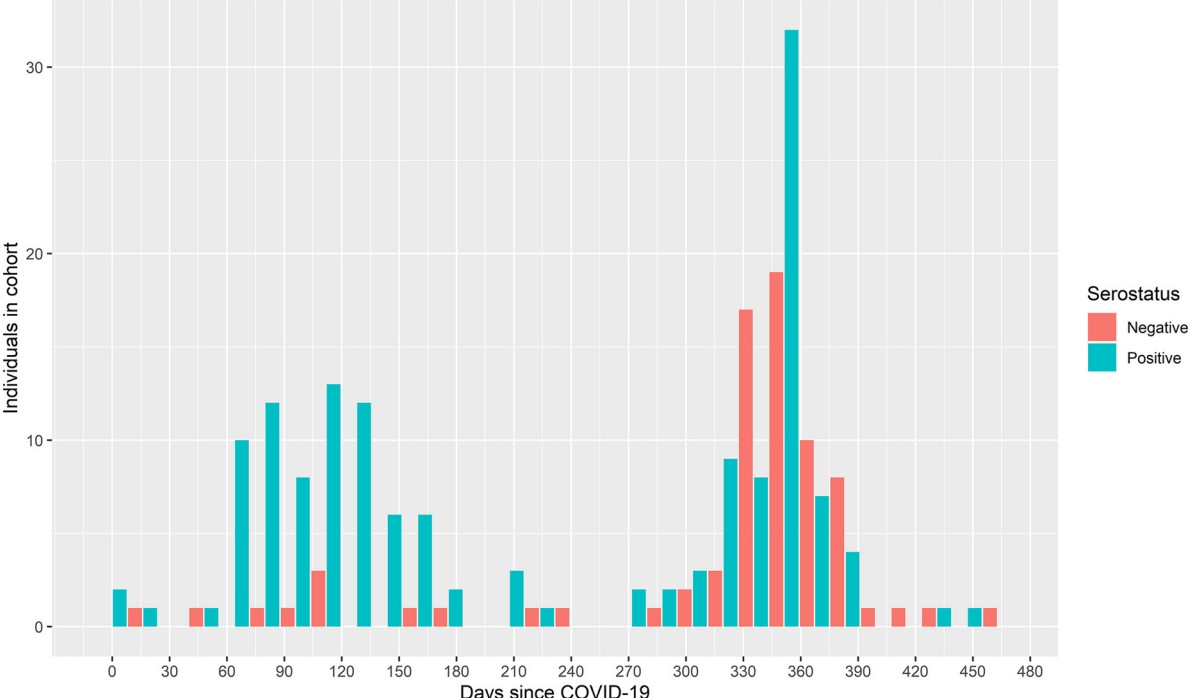

**FIG 2** Serostatus and time since COVID-19.

**Previous antibody tests and influenza vaccination.** In the group of seropositive individuals, 46.6% reported a prior positive SARS-CoV-2 antibody test, while the proportion was 20.2% in the group of seronegative individuals. This difference in previous antibody test results was statistically significant ($P < 0.001$).

The target group for influenza vaccination was expanded in the influenza season of 2020 to include the entire adult population, due to the COVID-19 pandemic and the fear that an extra burden of influenza could overpower hospitals. The frequency of individuals vaccinated in 2020 against influenza was similar in seropositive (34.1%) and seronegative (27.5%) participants ($P = 0.28$). The total number of individuals specifying that they received vaccination against influenza in 2020 (32%) was higher than in 2019 (19.7%), which was a significant increase ($P < 0.05$).

## DISCUSSION

The number of studies on long COVID is increasing rapidly, but studies based on serostatus are still few. We included 341 participants from a social media group of long-COVID sufferers who were tested for SARS-CoV-2 antibodies (antinucleocapsid) and subsequently completed an electronic questionnaire. The study was conducted in February and March 2021 right after the second surge of the COVID-19 pandemic in Denmark. The main findings can be summarized as follows: 90.6% of participants were women; 32% of the participants did not have detectable anti-nucleocapsid antibodies, indicating that their symptoms were most likely not related to COVID-19; seronegative

**TABLE 2** Logistic regression results based on long-COVID symptoms, age, sex and time since assumed COVID-19[a]

| Predictive variable | OR | 95% CI |
| --- | --- | --- |
| Age | 1.018 | 0.994–1.043 |
| Sex (female) | 0.878 | 0.345–2.119 |
| Time since assumed COVID-19 (days) | 0.987 | 0.984–0.991 |
| Sum of long-COVID symptoms | 1.000 | 0.999–1.001 |

[a]Area under curve, 0.79. McFadden, 0.2.

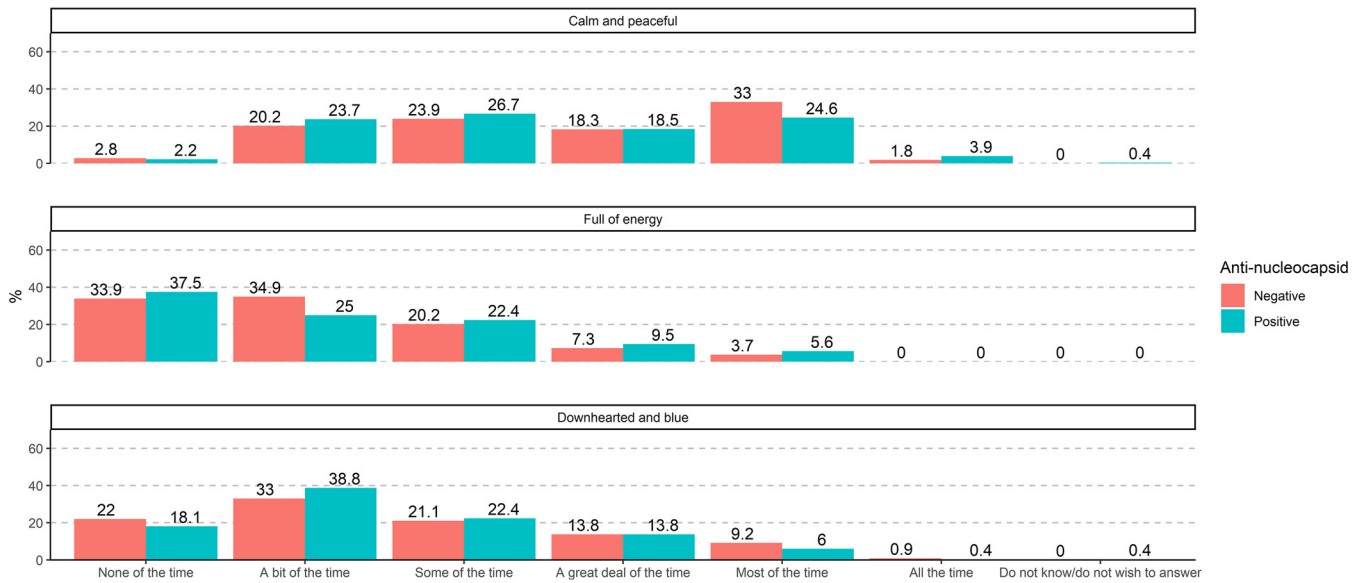

**FIG 3** Feelings in the previous 4 weeks.

individuals reported the longest time since COVID-19; and similar symptom burden was observed in the seropositive and the seronegative group.

The SARS-CoV-2 genome encodes four structural proteins, including spike (S), nucleocapsid (N), envelope, and membrane proteins. Development of IgG against the S protein is believed to be a neutralizing antibody immune response and has been the primary target for COVID-19 vaccines; therefore, detection of antibody responses against N protein, which plays a vital role in viral transcription and assembly, is more useful for distinguishing between serological responses to infection and vaccination (13, 14). Studies have shown long persistent humoral responses with detectable levels of IgG antibodies recognizing the N protein or the receptor binding domain (RBD) of the S protein at 8 months (15) and up to 12 months after symptom onset (16). As our study includes analysis for antibodies against N protein, we were able to examine natural infection rather than a potential vaccine response and to compare the positive and negative serology test results with the prevalence of symptoms.

**Serostatus and time since COVID-19.** For the group of seropositive individuals, time since COVID-19 was dispersed, with one peak around 3 to 4 months after infection and a second peak around 12 months after infection (Fig. 2). These peaks are in line with the pandemic surges in Denmark; the first surge was present 1 year before this study, and the second surge started 4 months before this study. Time since COVID-19 was longer for seronegative individuals ($P < 0.001$ in a univariate analysis) (Fig. S1). Knowing that the humoral immune response against SARS-CoV-2 recedes after natural infection as well as after vaccination, some of the individuals in the seronegative group could have had COVID-19 without detectable antibodies to prove it at the time of sampling. The possibility that they could have had positive serology if the blood had been sampled closer to the time of their COVID-19 infection cannot be ruled out. In a multivariate data analysis, the odds of having a positive serology sample decreased by 32.5% for each month that elapsed since perceived COVID-19.

**Long-COVID risk factors.** A recent systematic review (preprint) (17) and a rapid review by the National Institute for Health and Care Excellence (NICE) (1) found that female sex increases the risk of developing persistent symptoms after initial COVID-19 infection. Slower recovery in females has also been reported (18).

In this study, 90.6% of participants were women. Members of the social media group for long COVID sufferers were therefore more likely to be women than men (unless participation bias alone can explain this large difference), and no statistically significant

difference was found between sex and serostatus. It is not known why females are at higher risk of long-COVID symptoms, and more research is needed in this area.

**Long-COVID-related symptoms.** Vomiting and diarrhea were reported as being better after suspected or confirmed COVID-19 by the participants; all other symptoms surveyed were reported as being worse, regardless of serostatus. The seasonality of respiratory tract infections like COVID-19 is similar to the seasonality of viral gastroenteritis, which may explain why participants felt their burden of vomiting and diarrhea to be lower in the time following their assumed COVID-19 infection. Worsened sensitivity to sound was the only symptom significantly worse in the group of seropositive individuals compared to seronegative (Fig. 1) in a univariate analysis. Sensory deficits associated with COVID-19 have been observed in both the acute and post-acute phases, and one study has demonstrated that adult human inner ear tissue cells coexpress the angiotensin-converting enzyme 2 (ACE2) receptor and cofactors required for SARS-CoV-2 virus entry (19).

Having persistent symptoms may lead to the belief in having had COVID-19, especially in the context of a growing concern regarding long COVID, in spite of having no previous positive PCR or antibody test and in spite of the fact that other diseases may underlie symptoms attributed to COVID-19 infection. The belief in having had COVID-19 infection may have increased the likelihood of symptoms, either by directly affecting perception or indirectly by prompting maladaptive health behaviors (20).

**Health impact.** Neurocognitive symptoms have been found to persist for up to 1 year after COVID-19 symptom onset with psychiatric sequelae in the form of depression, anxiety, and cognitive problems (18). In a preprinted Danish study examining post-acute-phase symptoms and self-reported health problems 6 to 12 months following SARS-CoV-2 infection, new onset of mental and physical exhaustion was markedly overrepresented among individuals with previous SARS-CoV-2 infection compared to controls (18). Other post-acute-phase symptoms overrepresented among previous test-positive individuals were difficulties concentrating, memory issues, sleep problems, chronic fatigue syndrome, depression, and anxiety, as well as physical symptoms like taste and smell disturbance, fatigue, and dyspnea. Studies with long follow-up times are few but have found comparable patterns of persisting symptoms (21). This study showed a similar pattern, with a high proportion of participants reporting lack of energy and feeling downhearted and blue a bit of the time or more frequently (Fig. 3). This result, though, was not compared to a control group of the general population. We therefore do not know what the baseline levels of these symptoms were in the overall population at the time the study was conducted and whether this cohort differed significantly from them.

**Strengths and limitations of the study.** The main strength of this study is that the electronic questionnaire was completed by participants before the result of SARS-CoV-2 antibodies was given; therefore, serostatus did not affect the answers. Many studies are based on prior PCR results, but in this study, participants were included regardless of prior PCR results. Blood samples were analyzed for antibodies against nucleocapsid antigen, whereby we could differentiate whether participants were seropositive due to vaccination or natural infection.

Some weaknesses of this study should be mentioned. Comparison of smaller groups within the study cohort gives rise to possible misinterpretation, since the statistical power necessary to draw accurate conclusions about possible correlations might not have been achieved in these cases. Another limitation is that self-reporting of symptoms could be subject to recall bias, reporting bias, and misclassification, due to the retrospective study design and a long recall period or even long-COVID cognitive sequelae. Participants were from a social media group of long-COVID sufferers, so symptoms that impact daily life would most likely be remembered. Symptoms discussed within the social media group would also most likely be remembered. There is also a possibility of over- or underestimation of the reported associations; people living or working near the three hospitals where blood sampling took place may be overrepresented due to easier logistics. As the survey was distributed in an online social media group and most of the participants were women, participation bias cannot be ruled out. Written information was available only in Danish, and only participants who

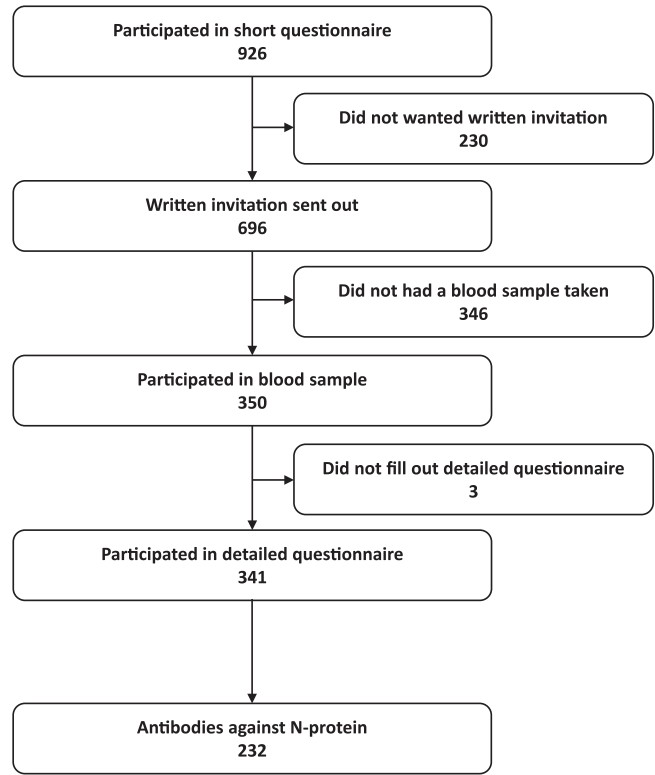

**FIG 4** Consort flow diagram.

were part of the social media group at the time the survey was performed were able to participate. The study does not include patient registry data, so results cannot be adjusted for severity of illness or comorbidity.

**Conclusion.** Our study confirms that long-COVID sufferers are mostly women. A third of the participants did not have detectable anti-N-protein antibodies, and seronegative individuals reported the longest time since COVID-19. Also, a similar symptom burden was observed in the seropositive and the seronegative groups. To conclude, it is important to verify COVID-19 early, as this study indicates an overlap between long-COVID symptoms and symptoms that are possibly of another origin.

## MATERIALS AND METHODS

**Study period: the COVID-19 pandemic in Denmark.** The study period spans a year from February 2020 to March 2021, including both the first and second surge of the COVID-19 pandemic in Denmark, when the index and Delta variants were observed. In November 2020, the Danish Health Authority recommended establishment of regional clinics/departments with the specific task of treating and rehabilitating citizens suffering from long COVID, so-called "long-COVID clinics" (22). The first Danish individuals were vaccinated in December 2020 (23), and test activity changed significantly over the course of the study period. In May 2020, free access to SARS-CoV-2 testing became available for all adults in Denmark (24). Test activity has fluctuated depending on the season, the rising vaccination status in 2021, and the requirements for a COVID pass in public venues (25).

**Study design and participation.** In this retrospective cohort study, all 8,000 members of the social media group Facebook "Covidramte med senfølger" ("long COVID sufferers") above the age of 17 years and living in Denmark were invited to participate in a short electronic questionnaire about long COVID, of which 926 participated. In the questionnaire, participants were offered further written information about this observational cohort study; 696 participants consented to receive a written invitation by email.

Three hundred fifty of the invited participants came to the hospital and had a blood sample taken at one of three geographical locations distributed across Denmark, i.e., Herlev, Odense, and Aarhus, between 10 February and 11 March 2021. In addition to blood sampling, participants were asked to complete an electronic questionnaire regarding symptoms, demographics, self-reported previous SARS-CoV-2 test results, vaccination, and physical and mental health. Of the 350 participants, 341 answered this more in-depth questionnaire (Fig. 4). The participants did not know the results of their blood samples before filling in the questionnaire. Answers to the questionnaire were registered in Research Electronic Data Capture (REDCap), a secure web-based tool for creating and managing surveys and data.

**Detection of antibodies against the nucleocapsid antigen.** Blood samples were analyzed for SARS-CoV-2 nucleocapsid (N) protein, as a proxy for natural SARS-CoV-2 infection, as some participants could have received one or more SARS-CoV-2 vaccines prior to blood sampling, which would result in the formation of antibodies against the spike protein but not the N protein. The presence of SARS-CoV-2 N-protein antibodies was determined by electrochemiluminescence (Elecsys anti-SARS-CoV-2 assay; Roche Diagnostics, GmbH, Germany) and a COBAS analyzer system (Roche Diagnostics) according to the manufacturers' instructions (26). A positive immune response was defined by a cutoff index (COI) of $>1$, with a specificity of 99.8%.

**Outcome measures.** The primary outcome of interest was to explore the proportion of the study population with antibodies against the nucleocapsid antigen (as a proxy for previous infection) and the association with supposed long-COVID symptoms.

**Approvals, ethics, and registrations.** The study was approved by the regional ethics committee of the Capital Region of Denmark (H-20072862), complied with the Helsinki II declaration, and was registered with the Danish data protection authorities (P-2020-1208). Participation was voluntary. Following the guidelines for providing oral and written information, all participants gave written informed consent to participate in the study. All personal data obtained in REDCap were kept in accordance with the general data protection regulation and data protection law stated by the Danish Data Protection Agency.

**Statistical analysis.** Univariate analysis was performed on baseline characteristics of seropositive compared to seronegative individuals; these are presented as numbers and percentages for categorical values. Continuous values are presented as medians and interquartile ranges or means and standard deviations (SD). The Wilcoxon rank sum test and chi-square test were used for comparisons of groups for continuous and categorical values, respectively. Participants specified their degree of persistent symptoms after supposed COVID-19 compared to that before supposed infection as a number between $-50$ and 50, with 0 being no change in symptoms, 50 being "much better," and $-50$ being "much worse." Therefore, a mean for each symptom was calculated for both groups (seronegative and seropositive). $P$ values were calculated to assess whether seronegative and seropositive individuals differed significantly in their burden of long-COVID symptom.

$P$ values of $<0.05$ were considered statistically significant.

Logistic regression was used to assess the relationship between serostatus and long-COVID symptoms while also adjusting for relevant independent variables (age, sex and time since COVID). Whenever values of age or sex were missing, the row was excluded ($n = 3$). Other missing values were handled by generating 10 imputed data sets using predictive mean matching. Parameters were analyzed for each imputed data set individually, and results were pooled (according to Rubin's rule). Collinearity was assessed by estimating the variance inflation factor (VIF) within the individual imputed data sets and, when relevant, omitting one of the correlated variables from the final model. Examples of highly correlated variables were the long-COVID symptoms vomiting and nausea. Goodness of fit was measured by testing a model for each imputed data set using the Hosmer-Lemeshow test ($P$ values were between 0.23 and 0.4). An initial, overfitted model included all long-COVID symptoms, in order to explore if certain symptoms showed predictive value. Afterward, a sum of long-COVID symptoms was calculated for each person within each imputed data set, and the means from these were used instead of the individual long-COVID symptoms. Results of the final model are presented as odds ratios (OR) and associated 95% CI.

Data management, statistical analyses, and figures were performed and created using R version 3.6.1.

## SUPPLEMENTAL MATERIAL

Supplemental material is available online only.

**SUPPLEMENTAL FILE 1**, PDF file, 0.1 MB.

## ACKNOWLEDGMENTS

We thank all participants who voluntarily contributed to this study and those involved in recruitment and sample processing. We also thank Lisbeth Andreasen from the Department of Clinical Biochemistry at Rigshospitalet for her technical assistance in analyzing samples for N-protein antibodies.

This work was financially supported by grants from the Carlsberg Foundation (CF20-4760045, to P.G.) and the Novo Nordisk Foundation (NFF205A0063505 and NNF20SA0064201, to P.G.). The funding source of this study did not influence the study design, data collection, data analysis or reporting.

The study was designed and initiated by K.F. and K.I. Data collection was done by K.F., R.B.H., A.R.R.E., K.I., I.S.J., and L.Ø. Sampling analyses were done by C.B.H., P.G., R.F.-S., and L.M.H. Statistical analysis and visualization were done by T.G.L. and K.F. K.F., T.G.L., R.B.H., and K.I. had full access to the data. The first draft of the manuscript was written by K.F., T.G.L., H.B., and K.I. K.F., T.G.L., C.B.H., R.B.H., A.R.R.E., R.F.-S., L.M.H., H.B., L.Ø., I.S.J., I.H., P.G., and K.I. critically revised the manuscript. All authors took part in conceptualization, interpretation, and discussion of results, and all agreed to be

accountable for all aspects of the work and approved the final version of the manuscript.

All authors declared no potential conflict of interest with respect to the research, authorship, and/or publication of this article.

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
