## [Reviewer comments · Microbiology Spectrum]

Microbiology Spectrum

Self-reported Long COVID and the association to SARS-CoV-2 antibodies in a Danish cohort up to 12 months after infection

Kamille Fogh, Tine Larsen, Cecilie Hansen, Rasmus Hasselbalch, Alexandra Eriksen, Henning Bundgaard, Ruth Frikke-Schmidt, Linda Hilsted, Lars Østergaard, Isik Johansen, Ida Hageman, Peter Garred, and Kasper Iversen

Corresponding Author(s): Kamille Fogh, Herlev and Gentofte Hospital, University of Copenhagen

Review Timeline:

Submission Date:	July 4, 2022
Editorial Decision:	August 16, 2022
Revision Received:	September 10, 2022
Accepted:	October 2, 2022

Editor: Holly Ramage

Reviewer(s): Disclosure of reviewer identity is with reference to reviewer comments included in decision letter(s). The following individuals involved in review of your submission have agreed to reveal their identity: Michael Asamoah-Boaheng (Reviewer #1); Ghulam Abbas (Reviewer #2); Nabil Benazi (Reviewer #3)

Transaction Report:

DOI: <https://doi.org/10.1128/spectrum.02537-22>

August 16, 2022

Dr. Kamille Fogh
Herlev and Gentofte Hospital, University of Copenhagen
Dept. of Cardiology & Dept. of Emergency Medicine
Herlev
Denmark

Re: Spectrum02537-22 (Self-reported Long COVID and the association to SARS-CoV-2 antibodies in a Danish cohort up to 12 months after infection)

Dear Dr. Kamille Fogh:

Thank you for submitting your manuscript to Microbiology Spectrum. As you will read, your manuscript was evaluated by two reviewers. While the reviewers found the work interesting, they both raised concerns regarding statistical analyses used to evaluate the data. Therefore, we invite you to submit a revised manuscript that addresses these concerns. When submitting the revised version of your paper, please provide (1) point-by-point responses to the issues raised by the reviewers as file type "Response to Reviewers," not in your cover letter, and (2) a PDF file that indicates the changes from the original submission (by highlighting or underlining the changes) as file type "Marked Up Manuscript - For Review Only". Please use this link to submit your revised manuscript - we strongly recommend that you submit your paper within the next 60 days or reach out to me. Detailed instructions on submitting your revised paper are below.

Link Not Available

Sincerely,

Holly Ramage

Journals Department
Reviewer comments:

Reviewer #1 (Comments for the Author):

The author investigated the association between self-reported long COVID and SARS-CoV-2 antibodies in a Danish cohort 12 months after SARS-CoV-2 infection. The research is interesting and it was well written. I must commend the authors for a good job done.

Although, this piece of work is great and interesting, i have some concerns that needs to be addressed:

It would be great for the authors to run a binary logistic regression with their data. Where the outcome variable will be the "serostatus" against the multiple long COVID-19 symptoms (as the exposure variables), while controlling/adjusting for some of the baseline factors summarized in Table 2 including "time since perceived COVID and blood sampling". This will make the results of this study stronger and more robust. The multivariate logistic regression will also help address some of your study limitations.

METHODS

Also, I would like to know the sampling technique the researchers used for the survey via the social media platforms. This will enable the researcher or the reviewer to have an idea about whether the participants sampled were from similar geographical location or school or family etc. If such were found, these factors would need to be blocked or controlled. In so doing, a mixed effect logistic regression model will be appropriate instead of a normal logistic regression for your multivariate analysis.

Reviewer #2 (Comments for the Author):

Taking an interest in the evolution of long-COVID as well as these risk factors is an innovative and very important idea because covid-19 hides many surprises concerning its evolution and especially these complications.

In your study you proceeded with a questionnaire made up of more than 14 variables, whose judgment criterion, i.e. the variable explained (response variable) is "The primary outcome of interest was to explore the proportion of the study population with antibodies against the nucleocapsid antigen and the association to supposed Long COVID" which is an objective and robust endpoint.

Nevertheless, any scientific study of these results must be validated by adequate statistical tests or even adequate and robust statistical models to be judged clear and valid.

In your study, section methods, paragraph "Statistical analysis", you carried out a univariate analysis of the basic characteristics of HIV-positive compared to HIV-negative individuals. And you used Wilcoxon rank sum statistical tests and the chi-square test were used for group comparisons for continuous and categorical values, respectively. All this with the aim of bringing out a statistically significant association between your explanatory variables and the response variable (positive serology).

The univariate analysis does not take into consideration the relationships between your explanatory variables such as collinearity between two or more explanatory variables or the phenomenon of confusion of the variables which must be controlled and verified in each model chosen and must be mentioned in the results of the study, whether for a single patient (individual) or between individuals. This is why the Anglo-Saxons invented the multivariate analysis using different regression models according to the requirement of the study, alongside them the French school which invented the multivariate analysis much more descriptive such as ACP, AFC, ...

However, in your study you only used univariate tests, which makes your study less robust and invalid from a statistical point of view and in this context I suggest that you review the statistical concept in this study and choose the most appropriate model. appropriate in order to bring out a significant relationship between the response variable and your explanatory variables.

Staff Comments:

Preparing Revision Guidelines

Please return the manuscript within 60 days; if you cannot complete the modification within this time period, please contact me. If you do not wish to modify the manuscript and prefer to submit it to another journal, please notify me of your decision immediately so that the manuscript may be formally withdrawn from consideration by Microbiology Spectrum.

The author investigated the association between self-reported long COVID and SARS-CoV-2 antibodies in a Danish cohort 12 months after SARS-CoV-2 infection. The research is interesting and it was well written. I must commend the authors for a good job done.

Although, this piece of work is great and interesting, i have some concerns that needs to be addressed:

It would be great for the authors to run a binary logistic regression with their data. Where the outcome variable will be the "serostatus" against the multiple long COVID-19 symptoms (as the exposure variables), while controlling/adjusting for some of the baseline factors summarized in Table 2 including "time since perceived COVID and blood sampling". This will make the results of this study stronger and more robust. The multivariate logistic regression will also help address some of your study limitations.

METHODS

Also, I would like to know the sampling technique the researchers used for the survey via the social media platforms. This will enable the researcher or the reviewer to have an idea about whether the participants sampled were from similar geographical location or school or family etc. If such were found, these factors would need to be blocked or controlled. In so doing, a mixed effect logistic regression model will be appropriate instead of a normal logistic regression for your multivariate analysis.

6. September 2022

To Editor

Holly Ramage

Microbiology Spectrum

Dear Holly Ramage

Thank you for the opportunity to revise and improve our manuscript

“Self-reported Long COVID and the association to SARS-CoV-2 antibodies in a Danish cohort up to 12 months after infection”.

We appreciate the time and effort that the reviewers have spent on the revision of our manuscript, and we have carefully addressed all comments and concerns from the reviewers in the point-by-point reply below.

We consider the paper much improved and hope that you will consider it for publication in **Microbiology Spectrum**.

Sincerely yours

Kamille Fogh, MD, Ph.d. student

Department of Cardiology and Department of Emergency Medicine

Herlev-Gentofte Hospital

Borgmester Ib Juuls Vej 1

DK - 2730 Herlev

T +45 2679 8310

E kamille.fogh.01@regionh.dk

Placement of revision (highlighted in red) in each response refers to placement in clean version.

Reviewer #1:

Statistical analysis

1. It would be great for the authors to run a binary logistic regression with their data. Where the outcome variable will be the "serostatus" against the multiple long COVID-19 symptoms (as the exposure variables), while controlling/adjusting for some of the baseline factors summarized in Table 2 including "time since perceived COVID and blood sampling". This will make the results of this study stronger and more robust. The multivariate logistic regression will also help address some of your study limitations.

Response to Reviewer, comment 1:

Thank you for this very relevant suggestion. We have done a binary logistic regression, which we found was the most suited analysis to run on our data and have added a section on this to both the "methods" section and the "results" section.

Prior to revision:

Method section - Statistical analysis:

Univariate analysis was performed on baseline characteristics of seropositive compared to seronegative individuals, these are presented as numbers and percentages for categorical values. Continuous values are presented as medians and interquartile ranges or means and standard deviations (SD). The Wilcoxon rank sum test and chi-square test were used for comparisons of groups for continuous and categorical values, respectively. Participants specified their degree of persistent symptoms after supposed COVID-19 compared to before supposed infection, as a number between -50 and 50: 0 being no change in symptoms, 50 being "much better", and -50 being "much worse". Therefore, a mean for each symptom was calculated for both groups (seronegative and seropositive). P-values were calculated to assess whether seronegative and seropositive individuals differed significantly in their burden of Long COVID symptom. P-values < 0.05 were considered statistically significant. Data management, statistical analyses, and figures were performed and created using R version 3.6.1.

Result section – Time since COVID-19 infection

Figure 2 and Supplementary figure 1 show the time since the assumed first day of COVID-19 for seronegative and seropositive individuals. Four individuals were excluded from this part of the analysis; three because they specified a date that was after the date they filled out the questionnaire, and one because the individual specified the same date as they filled out the questionnaire. Time since COVID-19 was significantly longer for the group of seronegative ($p < 0.001$) with a narrow interquartile range compared to the group of seropositive (see supplementary figure 1). Mean time since COVID-19 was 248.9 days (95%CI = 234.9-262.9) for the entire cohort, 317 days (95%CI = 306.7 – 327.3) for seronegative and 213.9 days (95%CI = 199.6 – 228.2) for seropositive. Figure 2 is a histogram showing the dispersion of the "time since the first day of assumed COVID-19 infection" within the cohort. Two peaks appear, the first one around 3-4 months after COVID-19

which is dominated by seropositive individuals. The second peak is around one year after infection and consists of both seronegative and seropositive individuals.

Revision (page 6, l. 82-94, page 8, l. 116-140 and a new table 2):

Method section - Statistical analysis:

Univariate analysis was performed on baseline characteristics of seropositive compared to seronegative individuals, these are presented as numbers and percentages for categorical values. Continuous values are presented as medians and interquartile ranges or means and standard deviations (SD). The Wilcoxon rank sum test and chi-square test were used for comparisons of groups for continuous and categorical values, respectively. Participants specified their degree of persistent symptoms after supposed COVID-19 compared to before supposed infection, as a number between -50 and 50: 0 being no change in symptoms, 50 being “much better”, and -50 being “much worse”. Therefore, a mean for each symptom was calculated for both groups (seronegative and seropositive). P-values were calculated to assess whether seronegative and seropositive individuals differed significantly in their burden of Long COVID symptom. P-values < 0.05 were considered statistically significant.

Logistic regression was used to assess the relationship between serostatus and Long-COVID symptoms while also adjusting for relevant independent variables (age, sex and “time since COVID”). Whenever values of age or sex were missing, the row was excluded (n = 3). Other missing values were handled by generating 10 imputed datasets using predictive mean matching. Parameters were analyzed for each imputed dataset individually and results pooled (according to Rubin’s Rule). Collinearity was assessed by estimating the VIF (variance inflation factor) within the individual imputed datasets and, when relevant, omitting one of the correlated variables from the final model. Examples of highly correlated variables were the Long-COVID symptoms “vomiting” and “nausea”. Goodness-of-fit was measured by testing a model for each imputed dataset using Hosmer-Lemeshow’s test (p-values were between 0.23 and 0.4). An initial, overfitted model included all Long-COVID symptoms, in order to explore if certain symptoms showed predictive value. Afterwards, a “sum of Long-COVID symptoms” was calculated for each person within each imputed dataset and the mean from these used instead of the individual Long-COVID symptoms. Results of the final model are presented as Odds Ratios (OR) and associated 95% Confidence Intervals (95% CI). Data management, statistical analyses, and figures were performed and created using R version 3.6.1.

Result section – Long COVID related symptoms and time since COVID-19 infection

All symptoms were reported to be worse after than before assumed COVID-19 for both the seropositive and the seronegative group. Exceptions to this were the symptoms “vomiting” and “diarrhea”, which were reported as “better” after assumed COVID-19 (figure 3). Only “sensitivity to sound” was significantly worse in the group of seropositive compared to seronegative (p < 0.05), all other symptoms were not significantly different. Figure 2 and Supplementary figure 1 show the time since the assumed first day of COVID-19 for seronegative and seropositive individuals. Four individuals were excluded from this part of the analysis; three because they specified a date that was after the date they filled out the questionnaire, and one because the individual specified the same date as they filled out the questionnaire. Time since COVID-19 was significantly longer for the group of seronegative (p < 0.001) with a narrow interquartile range compared to the group of seropositive (see supplementary figure 1). Mean time since COVID-19 was 248.9 days (95%CI = 234.9-262.9) for the entire cohort, 317 days (95%CI = 306.7 – 327.3) for seronegative and 213.9 days (95%CI = 199.6 – 228.2) for seropositive. Figure 2 is a histogram showing the dispersion of the “time since the first day of assumed COVID-19 infection” within the cohort. Two peaks appear, the first one around 3-4 months after

COVID-19 which is dominated by seropositive individuals. The second peak is around one year after infection and consists of both seronegative and seropositive individuals.

The individual Long-COVID symptoms were not predictive for serostatus in an initial, overfitted logistic regression model where all symptoms were included as well as age, sex and “time since assumed COVID-19” (table 2). In the final model, symptom burden was pooled into a “sum of all Long-COVID symptoms” which still had no significant predictive value for serostatus. However, in the final model, a one day increase in “time since assumed COVID-19” was significantly associated with odds of a positive antibody sample decreasing by 1.3%. Therefore, a one month (30 days) increase in “time since assumed COVID-19” was associated with a reduction in the odds by 32.5%.

Table 2: Logistic regression results based on Long-COVID symptoms, age, sex and time since assumed COVID-19

Predictive variables	OR	95% CI
Age	1.018	0.994 – 1.043
Sex (female)	0.878	0.345 – 2.119
Time since assumed COVID-19 (days)	0.987	0.984 – 0.991
Sum of Long-COVID symptoms	1.000	0.999 – 1.001
Area under curve: 0.79		
McFadden: 0.2		

Method

2. Also, I would like to know the sampling technique the researchers used for the survey via the social media platforms. This will enable the researcher or the reviewer to have an idea about whether the participants sampled were from similar geographical location or school or family etc. If such were found, these factors would need to be blocked or controlled. In so doing, a mixed effect logistic regression model will be appropriate instead of a normal logistic regression for your multivariate analysis.

Response to Reviewer, comment 2:

Thank you for this comment.

All invited participants were from the same social media (Facebook) group with the possibility of broad national participation, as participants could live anywhere in Denmark. The intention was not to sample participants from similar geographical location or from the same family, workplace etc. Participants were only sampled once. For this reason, we choose to run a normal logistic regression on our data.

Reviewer #2:

Statistical analysis

1. In your study, section methods, paragraph "Statistical analysis", you carried out a univariate analysis of the basic characteristics of HIV-positive compared to HIV-negative individuals. And you used Wilcoxon rank sum statistical tests and the chi-square test were used for group comparisons for continuous and categorical values, respectively. All this with the aim of bringing out a statistically significant association between your explanatory variables and the response variable (positive serology).

The univariate analysis does not take into consideration the relationships between your explanatory variables such as collinearity between two or more explanatory variables or the phenomenon of confusion of the variables which must be controlled and verified in each model chosen and must be mentioned in the results of the study, whether for a single patient (individual) or between individuals. This is why the Anglo-Saxons invented the multivariate analysis using different regression models according to the requirement of the study, alongside them the French school which invented the multivariate analysis much more descriptive such as ACP, AFC, ...

However, in your study you only used univariate tests, which makes your study less robust and invalid from a statistical point of view and in this context I suggest that you review the statistical concept in this study and choose the most appropriate model. appropriate in order to bring out a significant relationship between the response variable and your explanatory variables.

Response to Reviewer, comment 1:

Thank you for pointing this out. We have now done a binary logistic regression and have added a section on this to both the "methods" section and the "results" section. See revisions in the answers above.

October 2, 2022

Dr. Kamille Fogh
Herlev and Gentofte Hospital, University of Copenhagen
Dept. of Cardiology & Dept. of Emergency Medicine
Herlev
Denmark

Re: Spectrum02537-22R1 (Self-reported Long COVID and the association to SARS-CoV-2 antibodies in a Danish cohort up to 12 months after infection)

Dear Dr. Kamille Fogh:

Your manuscript has been accepted, and I am forwarding it to the ASM Journals Department for publication. You will be notified when your proofs are ready to be viewed.

Sincerely,

Holly Ramage
Editor, Microbiology Spectrum
